# Hybrid Ionic Silver and Magnetite Microgels Nanocomposites for Efficient Removal of Methylene Blue

**DOI:** 10.3390/molecules24213867

**Published:** 2019-10-26

**Authors:** Ayman M. Atta, Amany K. Gafer, Hamad A. Al-Lohedan, Mahmood M. S. Abdullah, Ahmed M. Tawfeek, Abdelrahman O. Ezzat

**Affiliations:** 1Surfactant research chair, Chemistry department, college of science, King Saud University, Riyadh 11451, Saudi Arabia; 2Petroleum Application Department, Egyptian Petroleum Research Institute, Nasr City 11727, Cairo, Egypt

**Keywords:** water purification, cationic dye, adsorbents, Fenton oxidation catalyst, magnetite, silver nanocomposites

## Abstract

The ionic crosslinked 2-acrylamido-2-methylpropane sulfonic acid-co-acrylic acid hydrogel, AMPS/AA and its Ag and Fe_3_O_4_ composites were synthesized using an in situ technique. The surface charge, particle sizes, morphology, and thermal stability of the prepared AMPS/AA-Ag and AMPS/AA-Fe_3_O_4_ composites were evaluated using different analytical techniques and their adsorption characteristics were evaluated to remove the methylene blue cationic dye, MB, from their aqueous solutions at optimum conditions. Also, the same monomers were used to synthesize AMPS/AA microgel and its Ag and Fe_3_O_4_ nanocomposites, which were synthesized using the same technique. The AMPS/AA-Fe_3_O_4_ nanocomposite was selected as conventional iron-supported catalyst due to the presence of both Fe(II) and Fe(III) species besides its magnetic properties that allow their easy, fast, and inexpensive separation from the aqueous solution. It was then evaluated as a heterogeneous catalyst for complete MB degradation from aqueous solution by heterogeneous Fenton oxidation. It achieved a high rate of degradation, degrading 100 mg L^−1^ of MB during a short time of 35 min as compared with the reported literature.

## 1. Introduction

Crosslinked hydrophilic polymers are classified according to their particle size, and macrogels (hydrogels), microgels, and nanogels have been widely used as adsorbents for water purification and desalination [1,2,3,4,5]. The chemical structures, crosslinking densities, and compositions of the gels as well as their sizes were important factors for increasing their adsorption performances [6,7,8]. It was also reported that the innovative combination of polymer hydrogel networks with micro- or nanoparticles also improved their performance in different application fields, including catalysis, drug delivery, medicine, water treatment, and environmental remediation [9,10,11]. The micro- and nanoparticles, including metals, metal oxides, and polymeric moieties, were used to provide superior functionality and produce reactive nanocomposites [12,13,14]. There is a relatively little reported progress for the applications of nanomaterials in the field of water treatments as compared with their applications in medicine and electronics. They have been applied as effective adsorbents, filters, disinfectants, and reactive reagents for water treatment [15,16]. The mixing of nanoparticles with the polymeric hydrogels may result in synergistic property enhancement for either polymer hydrogels or nanoparticles. One of the incorporation advantages is that the mechanical strengths of polymeric networks were improved by the insertion of nanoparticles. The dispersion of nanoparticles was also improved by capping or interacting with the polymeric matrices. The challenges in the application of microgels and nanogels particles in water treatment are based on their negative impacts on human health and the environment. These negative impacts could potentially be overcome by their incorporation with nanoparticles. The dispersion of nanoparticles into polymer networks is an important factor when designing multifunctional composites based on the synthesis techniques [17]. The nanoparticle and microparticle hydrogel composites were prepared either by mixing the particles with monomers during the crosslinking of monomers or by forming the particles after formation of the hydrogels using an in situ technique [18,19,20]. It is very important to produce “smart” materials having multifunctional and stimuli-responsive properties during the design of polymer composites. The smart gel composites were used to produce soft active materials that are responsive to photo and thermal stimuli as well as magnetic fields, external pressure, and the surrounding pH for allow for application in aggressive environments and environmental remediation systems [21,22,23,24,25]. The magnetic gel composites were widely used as environmentally friendly adsorbents to remove toxic organic and inorganic water pollutants. Among the various composites, magnetic ones can easily be reused and collected by applying an external magnetic field [26,27,28]. Moreover, antimicrobial gel composites based on silver, zinc oxides, and titania have been preferred for catalytic oxidation–reduction of toxins/removal of pollutants [29,30]. In our previous work, microgel silver and magnetite nanocomposites based on 2-acrylamido-2-methylpropanesulfonic acid (AMPS) and acrylamide were prepared for applications as an adsorbent for water pollutants. In the present work, ionic monomers based on AMPS and acrylic acid (AA) were selected to increase the interaction of the gel with metal ions (iron and silver) using an in situ technique for preparing different gel composites with different morphologies. Moreover, this work aims to use these composites as faster adsorbents and catalytic reductants of methylene blue basic dye than reported in our previous work [31]. The kinetic and thermodynamic studies of the adsorption process for the prepared materials is another goal of the present study.

## 2. Results

The AMPS and AA monomers were selected to prepare random ionic copolymer with increased AA content in their chemical composition according to their reactivity ratios measurements [32]. The crosslinked AMPS/AA copolymers were prepared in micro- or macroscale for use as a template to synthesize nanocomposites by applying the in situ technique as illustrated in Scheme 1. The AMPS/AA hydrogel was prepared by radical crosslinking polymerization as reported in the experimental section. Its microgel is prepared by using free radical emulsion polymerization in the presence of surfactants as illustrated in the experimental part. The presence of higher surfactant content as the emulsifier to form an oil-in-water (O/W) emulsion, which controls the size of the AMPS/AA networks to form microgels under a suitable concentration of *N*,*N*-methylenebisacrylamide (MBA) crosslinker [33]. The formation of microgels is affected by the interaction between surfactant and crosslinked copolymer particles, which is based on the chemical structure of monomers as well as the chain length of the surfactant. Hence, the extent of surface activity plays an important role in the interaction between the surfactant and the microgel during and after synthesis [34]. The incorporation of Ag and Fe cations with amide and sulfonic and carboxylic anions of AMPS/AA depends on their ability to form chemical coordination bonds as well as the degree of carboxylic and sulfonic ionization (Scheme 1).

### 2.1. Characterization of AMPS/AA Nanocomposites

The morphologies of the hydrogel composites AMPS/AA-Ag, AMPS/AA-Fe_3_O_4_, and AMPS/AA microgel and its related nanocomposites AMPS/AA-Fe_3_O_4_ and AMPS/AA-Ag were investigated through TEM and SEM micrographs, as presented in Figure 1a–e and Figure 2a–c, respectively. The surface morphologies of AMPS/AA-Ag, and AMPS/AA-Fe_3_O_4_ hydrogel (Figure 1a,b) confirm that the Ag and Fe_3_O_4_ nanoparticles are embedded into the gel networks. The morphologies of AMPS/AA microgel particles (Figure 1e) are strongly influenced after the incorporation of Ag nanoparticles (NPs) (Figure 1d) and Fe_3_O_4_ NPs (Figure 1c). The surface morphologies (Figure 2a–c) also show that the surface roughness of AMPS/AA microgel changed with the incorporation of Ag and Fe_3_O_4_ nanoparticles.

The thermal stability and inorganic contents of AMPS/AA hydrogel and its composites and AMPS/AA microgel and its nanocomposites are determined by thermogravimetric measurements (TGA). Figure 3a,b respectively demonstrate TGA thermograms in macrosized and microsized AMPS/AA composites. The degradation temperatures (DT; °C), temperature losses 10 wt.% (T_10%_; °C), degradation stages, and remained residual (R%; wt.% above 750 °C) are determined from thermograms and summarized in Table 1. The initial weight loss (region A; T < 100 °C) referred to the loss of small water amount (bound water; 5 wt.%) which attributed to the hydrophilicity of the AMPS/AA networks. The AMPS/AA, AMPS/AA-Ag, and AMPS/AA-Fe_3_O_4_ hydrogel composites lost 5 wt.% at temperatures of 170, 200, and 160 °C, respectively (Figure 3a). Meanwhile, the AMPS/AA, AMPS/AA-Ag, and AMPS/AA-Fe_3_O_4_ microgel composites lost 5 wt.% at temperature 150, 220, and 180 °C, respectively (Figure 3b). It is observed that the incorporation of Ag and Fe_3_O_4_ NPs reduces the weight losses in the second stage, confirming the strong interaction of silver or magnetite with the carboxylic groups of the network, which retards their decarboxylation at an early decomposition stage. The data of R% (residual mass remaining after heating of the samples to 750 °C; Table 1 and Figure 3a,b) were used to confirm the remaining Ag and Fe_3_O_4_ NPs.

The X-ray powder diffractograms (XRD) of AMPS/AA-Fe_3_O_4_ hydrogel, AMPS/AA-Fe_3_O_4_ microgel, AMPS/AA-Ag hydrogel, and AMPS/AA-Ag microgel nanocomposites were represented in Figure 4a–d, respectively. The crystalline structures of Ag and Fe_3_O_4_ nanoparticles were confirmed from the diffraction peaks of the AMPS/AA microgels XRD diffractograms (Figure 4b,d). The AMPS/AA hydrogels diffractograms (Figure 4a,c) show a semicrystalline peak at 2-theta from 15 to 20° that affect the appearance of the Ag or Fe_3_O_4_ nanoparticle peaks. These data allow for elucidation of the interaction of Ag or Fe_3_O_4_ nanoparticles with AMPS/AA networks [20,31].

The particle sizes and polydispersity index (PDI) for AMPS/AA microgel and its Ag and Fe_3_O_4_ nanocomposites are determined from DLS measurements in aqueous solutions at different pH, and are represented in Figure 5a–c. The data show that different sizes of AMPS/AA microgels (Figure 5a) form aggregates at below 100 nm, several hundred nm and above 10 µm at pH 7, and the sizes of the aggregates increase in acidic (pH 4) and basic (pH 10) aqueous media. The lower sizes of AMPS/AA particles below 100 nm, were highly dispersed in the basic medium and more aggregated in the acidic medium. These observations are attributed to the protonation of sulfonic and carboxylic groups of AMPS/AA at pH 4 and their complete ionization in the basic medium. The particle sizes of AMPS/AA-Fe_3_O_4_ and AMPS/AA-Ag nanocomposites are increased in the basic medium compared to the acidic medium, which is related to increased swelling of the ionic microgel in the basic medium. The DLS data of the nanocomposites AMPS/AA-Fe_3_O_4_ and AMPS/AA-Ag (Figure 5b,c) show that the dominant form of Ag NPs is in small aggregates of size several hundred nanometers greater than the Fe_3_O_4_ NPs. Moreover, the size of the aggregates varied significantly with pH value. The surface charges of the AMPS/AA microgel and their nanocomposites were determined from zeta potentials (mV) measurements in their aqueous solutions with different pH and summarized in Table 2. In this respect, the AMPS/AA microgel and their nanocomposites are well dispersed in aqueous phases at pH 7 and possess negative charges (Table 2). The AMPS/AA-Ag microgel composite is more negative than other microgels at pH 7. These data suggest that the presence of single Ag NPs outside the microgel network facilitates the deprotonation of sulfonic and carboxylic groups of AMPS/AA networks at pH values above 4 (i.e., less negative values). Meanwhile, the Fe_3_O_4_ NPs that interacted at the core of AMPS/AA networks (TEM and DLS data; Figure 1 and Figure 4) facilitate the deprotonation of sulfonic and carboxylic groups of AMPS/AA network at pH values below 4, as confirmed from more negative values in aqueous solutions at different pHs.

The magnetic characteristics of AMPS/AA-Fe_3_O_4_ hydrogel and microgel composites were investigated using vibrating sample magnetometer (VSM) at room temperature. Their magnetic hysteresis loops are represented in Figure 6a,b. The saturation magnetization (Ms; emu/g), remnant magnetization (Mr measures the amount of magnetization that remains after the magnetic field removal; emu/g), and coercivity (Hc measure of the reverse field required to drive the magnetization to zero after being saturated; G) were determined and are listed in Table 3. The data confirm that the magnetic characteristics of AMPS/AA-Fe_3_O_4_ microgel are stronger than its hydrogel composite. This means that the higher shielding effect of AMPS/AA hydrogel and its lower magnetite contents (TGA data Table 1) compared to its microgel nanocomposite result in a reduction in its magnetic properties [23].

### 2.2. Application of AMPS/AA Hydrogel and Its Ag and Fe_3_O_4_ Composites as Methylene Blue (MB) Adsorbent

The relation between removal efficiency (E%) and weight contents (mg) of AMPS/AA hydrogel and its Ag and Fe_3_O_4_ composites in the presence of 500 mg L^−1^ of MB is plotted in Figure 7. The data confirm that AMPS/AA hydrogel and its composites achieve higher MB removal efficiencies at 6 mg content. The contact times for adsorption 500 mg L^−1^ of MB using 6 mg adsorbent versus their adsorption capacities are plotted in Figure 8.

The optimum pH for adsorbing 500 mg L^−1^ of MB in the presence of AMPS/AA hydrogel and its composites (6 mg) is determined from plots in Figure 9.

The adsorption kinetics of the adsorbate (MB) in the presence of MPS/AA hydrogel and its composites as adsorbents were investigated to detect the nature of the adsorption process and whether it was physical, physicochemical, or chemical. The pseudo-first-order and pseudo-second-order models can be determined from the relations as
ln (q_e_ − q_t_) = ln q_e_ − k_1_t(1)
(t/q_t_) = [(1/k_2_q_e_^2^) + t (1/q_e_)](2)

The q_t_ and q_e_ are the adsorption capacity (mg g^−1^) of the adsorbents at different time t and equilibrium, respectively. The adsorption rate constant of pseudo-first-order, k_1_ (min^−1^), and pseudo-second order, k_2_ (g mg^−1^ min^−1^), can be determined from the slope of the straight-line plots represented in Figure 10a,b. The rate constant parameters k_1_, k_2_, q_e_, q_exp_, and correlation coefficients, R^2^_1_ or R^2^_2_ are determined and summarized in Table 4.

One of the most important parameters to produce an effective adsorbent is the ability to reuse it for several cycles. In this respect, the AMPS/AA-Fe_3_O_4_ microgel composite can be easily collected from the aqueous by using an external magnet as illustrated from the photo represented in Scheme 2. The relation of efficiency percentages versus cycle numbers was represented in Figure 11. The AMPS/AA-Fe_3_O_4_ and AMPS/AA-Ag reused for five cycles with similar MB desorption and adsorption data. The fifth cycle of AMPS/AA-Fe_3_O_4_ reduces the MB dye adsorption efficiency to 75 wt.% while, the removal efficiency of AMPS/AA-Ag remains steady for five cycles. These data confirm the excellent dispersion and bonding of Ag NPs with AMPS/AA networks more than Fe_3_O_4_ NPs.

### 2.3. Catalytic of AMPS/AA–Fe_3_O_4_ Microgel Nanocomposites

The complete thermal degradation of MB with AMPS/AA-Fe_3_O_4_ nanocomposite was carried out in the presence of H_2_O_2_ and HCl using Fenton oxidation (as reported in the experimental section). The catalytic degradation of MB dye was investigated by UV-Vis spectra as represented in Figure 12. The relation of the quotient of the residual MB concentration at time t (Ct) over the initial MB concentration (C0) and contact time, t (min), in the presence of AMPS/AA-Fe_3_O_4_ nanocomposite is plotted on a semilogarithmic scale and represented in Figure 13.

## 3. Discussion

The distribution of the Ag and Fe cations embedded on AMPS/AA networks facilitates the reduction of these cations to form Fe_3_O_4_ and Ag NPs using ammonia or NaBH_4_, respectively, as represented in Scheme 1. The Ag NPs are uniformly highly dispersed into AMPS/AA hydrogel networks (Figure 1a) more than Fe_3_O_4_ nanoparticles (Figure 1b) which indicates that the high affinity of iron cations allows them to combine with polymer networks [35] and, also, the ability of Ag cations to be tetrahedrally coordinated with the AMPS/AA networks as ligand [36]. Meanwhile, the incorporation of Fe_3_O_4_ NPs into AMPS/AA microgel particles leads to the formation of core/shell morphologies (Figure 1c) with an average diameter of 0.2 µm. The surface morphology data (Figure 2) indicate that the incorporation of Ag NPs leads to enhanced roughness of the microgel particles (Figure 1d and Figure 2b) with the formation of some discrete Ag NPs (not integrated into microgel templet) beside the hybrid particles. More rough surfaces are observed in case of AMPS/AA-Ag microgel (Figure 2b) than AMPS/AA microgel (Figure 2a) and AMPS/AA-Fe_3_O_4_ microgel (Figure 2c). These observations confirm that the AMPS/AA microgel has a maximum amount of inorganic materials that can be embedded into the microgel networks. The TEM micrographs indicate that the average sizes of Ag NPs ranged from 10 to 20 nm (Figure 1d) and does not change the AMPS/AA microgel dimension (Figure 1e). The visible dark dots related to Ag NPs (Figure 1a,d) are homogeneously distributed in the surface layer of AMPS/AA hydrogels and microgels. The white visible circles of AMPS/AA-Fe_3_O_4_ microgel (Figure 1c) represent the presence of AMPS/AA microgel without loading with Fe_3_O_4_ NPs. The low penetration of Ag or Fe_3_O_4_ NPs deeper into the AMPS/AA microgel is attributed to the heterogeneity of the crosslinked AMPS/AA microgel due to the different reactivity of AMPS, AA, and AMPS/AA towards MBA crosslinker [37]. The high reactivity of AA towards MBA more than AMPS during the emulsion crosslinking polymerization can form a hard-crosslinked shell and weakly crosslinked core that prevents the diffusion of Ag or Fe_3_O_4_ NPs into the deeper AMPS/AA microgel core [38]. The TGA thermograms (Figure 3 and Table 1) data indicate that the content of bound water decreased with the incorporation of nanoparticles in case of Ag more than Fe_3_O_4_ NPs. Moreover, the hydrophobicity of AMPS/AA networks increased in the microsizes more than macrosizes due to the increased crosslink density of networks at the exterior compared to the interior of microgels. The T_10%_ data (Table 1) confirm that the thermal stability of AMPS/AA microgel composites is higher than the hydrogels composites. The second decomposition stage (Table 1; from 100 to 350 °C) indicates the decarboxylation of the acrylic groups of the AMPS/AA networks. The data of the second stage (Table 1) indicate that more bonding of Ag and Fe_3_O_4_ NPs occur in AMPS/AA hydrogels than microgels, as confirmed from the TEM data (Figure 1a–e). In the third decomposition stage (from 350 to 450 °C), the thermal decomposition of the C-C backbone and complete AMPS/AA polymer degradation take place. The data of R% (Table 1) indicate the higher contents of Ag and Fe_3_O_4_ NPs embedded in AMPS/AA microgel composites than their hydrogel composites. Moreover, the TGA measurements clearly show a considerable improvement in the thermal properties for obtained AMPS/AA composite materials due to the organized distribution and bonding of Ag and Fe_3_O_4_ NPs in their networks. The presence of Ag and Fe_3_O_4_ NPs reduces the mobility of the AMPS/AA chains that will suppress the thermal degradation by minimizing the radical formation due to chain transfer reactions to improve their thermal stability [38,39]. The formation of magnetite without formation of other iron oxides and silver without silver oxides as evidenced by the XRD diffractogram diffraction peaks that appeared (Figure 4a–d) indicate that AMPS/AA networks act as good capping for Ag and Fe_3_O_4_ NPs to protect their oxidation from the surrounding environments [31]. The particle sizes of AMPS/AA-Fe_3_O_4_ and AMPS/AA-Ag nanocomposites are increased in the basic medium compared to the acidic medium, which is related to increased swelling of the ionic microgel in basic medium [40]. It was also observed that the AMPS/AA microgel disaggregates into fully dispersed primary particles with the incorporation of the Ag NPs (Figure 5c). No such dispersed particles were observed in the AMPS/AA-Fe_3_O_4_ microgel nanocomposites. TEM images (Figure 1a–e) confirm that single Ag NPs are formed in case of AMPS/AA-Ag microgel nanocomposites, while the TEM micrograph of AMPS/AA-Fe_3_O_4_ (Figure 1c) indicates the appearance of lower sizes of AMPS/AA nanogels which do not contain single Fe_3_O_4_ NPs. There is thus good agreement between TEM and DLS data of AMPS/AA microgel nanocomposites, although slight contradictions may have been due to sample polydispersity and different biases of the measurements involved. If the surface charge values of particles as suspension or dispersion are more negative or positive than 25 mV, it means that they are stable in aqueous solutions [41]. The outer surface charges of the colloids confirm the intramolecular and intermolecular interactions of the colloids with the surrounding environments [42]. The low negative surface charge values of AMPS/AA microgel nanocomposites in basic and acidic medium suggest that the formation of hard-crosslinked shell and weakly crosslinked core (TEM and SEM data; Figure 1 and Figure 2) prevents the deprotonation of sulfonic and carboxylic groups of AMPS/AA network even in basic aqueous solutions [43]. The data listed in Table 3 and Figure 6a,b indicate that the AMPS/AA-Fe_3_O_4_ microgel has highest Ms and lower Hc or Mr values than its hydrogel nanocomposite. The magnetization data of AMPS/AA-Fe_3_O_4_ nanocomposites (Figure 6 and Table 3) suggests their ability to respond to an external magnet for application as adsorbent or catalyst in water treatment and other environmental issues.

The AMPS/AA hydrogels nanocomposites are evaluated as an adsorbent to remove the MB from water due to their good bioactivity and magnetic properties that assist in collecting the adsorbent for reuse after being used for the removal of pollutants. The adsorption characteristics of AMPS/AA-Ag nanocomposite are evaluated as bioactive adsorbent as well as the high surface area of silver NPS along with the ionic character of AMPS/AA sulfonic and carboxylic groups. The optimum conditions, such as adsorbents contents, adsorbent contact times, and pH of MB solutions were investigated, as shown in Figure 7, Figure 8 and Figure 9, for application in AMPS/AA hydrogels and their composites as adsorbents. The data represented in Figure 7, Figure 8 and Figure 9 show that the contact times of AMPS/AA-Ag, AMPS/AA, and AMPS/AA-Fe_3_O_4_ composites to adsorb maximum concentration of MB were 24, 36, and 60 min, respectively. These data agree with that reported for the morphologies of AMPS/AA hydrogel composites (Figure 1a,b). The presence of uniformly highly dispersed Ag NPs into AMPS/AA networks, more so than with Fe_3_O_4_ nanoparticles, facilitates the interaction and diffusion of MB into AMPS/AA-Ag nanocomposite networks than magnetite composites [44]. It is noticed that the adsorption capacities of AMPS/AA increased with increasing pH of solutions up to pH 9 to reach stability. This behavior can be attributed to the protonation of sulfonic and carboxylic groups of AMPS/AA in acidic pH and deprotonation of sulfonic and carboxylic groups in the basic medium to increase the electrostatic attraction between the positive charges of MB and the negative charges of AMPS/AA in basic medium. It was also observed that the AMPS/AA-Ag nanocomposite had the highest MB removal at neutral pH 7 (Figure 9). The incorporation of Ag or Fe_3_O_4_ NPs into AMPS/AA networks reduces the MB removal efficiencies of the composites at basic conditions. This observation can be attributed to the strong interactions of nanoparticles with the ionic groups of AMPS/AA networks, which reduces their deprotonation in the basic medium [45]. The data (Table 4, Figure 10a,b) reveal that the linear plots and agreements between the experimental q_exp_ and q_e_ values for AMPS/AA and its composites are consistent with the pseudo-second-order kinetic model. The values of k_2_ indicate that the adsorption rate order can be arranged as AMPS/AA-Ag > AMPS/AA-Fe_3_O_4_ >AMPS/AA, and suggest that the physicochemical interaction mechanism is more favorable due to the presence of nanoparticles into composites [46]. The physical mechanism of AMPS/AA nanocomposites represented in Scheme 2 shows the electrostatic interactions between the negative charges of AMPS/AA and positive charges of MB. Moreover, the rough regular and spherical Ag NPs, besides their nanometer scale and relatively high specific surface area, are also responsible for the higher MB adsorption efficiency with AMPS/AA-Ag nanocomposites [25]. Accordingly, the possible adsorption mechanism includes the adsorption of MB and the interactions between MB and Fe_3_O_4_ or Ag NPs.

The solid particles of either AMPS/AA hydrogel or AMPS/AA-Ag composite are collected by filtration. The MB desorption experiments for AMPS/AA hydrogel and its composites were carried out in an acidic solution as reported in the experimental section. It is very important to investigate the effect of acid on solubility or destroying of nanoparticles which effect on their reusability as adsorbents. It is observed that the desorption of MB from AMPS/AA hydrogel and its composites was carried out in 15 and 45 min, respectively. The recycling data (Figure 11) showed the destruction of AMPS/AA hydrogel networks after three cycles of adsorption/desorption, and 65% removal efficiency was achieved in the fifth cycle (Figure 11). This result confirms the destruction of the electrostatic attraction between AMPS/AA composite and MB molecules.

The applications of Fe_3_O_4_ and Ag NPs for catalytic either photo or thermal degradation of toxic organic dyes has previously been reported [25,47,48]. The Ag NPs were used to change the oxidized form of MB (blue color) to the reduced form (colorless) without reduction in toxicity [25]. The oxidation of organic pollutants using magnetite leads to the formation of intermediate species, which can be further oxidized up to CO_2_, H_2_O, and (if the pollutant contains heteroatoms, such as MB dye) inorganic salts [47,48]. The data represented in Figure 12 show that MB was not degraded in the absence of AMPS/AA-Fe_3_O_4_ nanocomposite even after 24 h (Figure 10a). The presence of AMPS/AA-Fe_3_O_4_ nanocomposite completely degraded the MB without formation of an intermediate during 35 min (Figure 12b). The disappearance of any peak up to 200 nm indicates the complete degradation of MB without the formation of organic intermediates. These data suggest the higher activity of AMPS/AA-Fe_3_O_4_ as a catalyst when compared with previous work related to the application of iron oxides as a catalyst for dye removal [47,48,49]. The straight line with R^2^ value of 0.993 is obtained, and indicates (pseudo) first-order reaction kinetics [50]:ln(Ct/C0) = −k_obs_t(3)
where k_obs_ (min^−1^) is the observed degradation rate constant which has a value of 0.0263 per minute. The pseudo first order reaction kinetics confirms Fenton-like oxidation in which magnetite in the AMPS/AA-Fe_3_O_4_ microgel nanocomposite performed as a Fe(III) source [47,48,49]. The high degradation rate of the AMPS/AA-Fe_3_O_4_ microgel nanocomposite correlates well with the contents of Fe_3_O_4_ (25 wt.%; Table 1) having large specific surface area due to their lower particle sizes. Moreover, there is good dispersion of the AMPS/AA-Fe_3_O_4_ microgel nanocomposite in their aqueous solutions due to their negative surface charges (Table 2; more negative than 20 mV) and its relatively low crystallinity due to capping with AMPS/AA amorphous microgel.

The separation of magnetite with the external magnet after two cycles suggests that the magnetite NPS dissolves in the reaction medium to produce Fe(III) that enhances Fenton oxidation to cause more effective degradation of MB. Finally, it can be concluded that the catalytic characteristics of AMPS/AA-Fe_3_O_4_ are attributed to the adsorption of the outer surface of the Fe_3_O_4_ nanoparticles, which was confirmed by the lower absorbance of MB after 2 min (Figure 12a). The adsorption of MB followed by its degradation with the recycling between Fe(III) and Fe(II) at the particle surface is the rate-determining step (pseudo-first-order reaction kinetics Figure 13). Moreover, the iron redox state in the particle core plays only a minor role to determine the catalytic activity of AMPS/AA-Fe_3_O_4_.

## 4. Materials and Methods

### 4.1. Materials

All chemicals used in this work of analytical grade were purchased from Aldrich Chemicals Co. (Missouri, MO, USA) Acrylic acid (AA), 2-acrylamido-2-methylpropane sulfonic acid (AMPS), *N*,*N*-methylenebisacrylamide (MBA), Span-85 and ammonium persulfate (APS) were used to prepare AMPS/AA crosslinked microgel. The reagents AgNO_3_ FeCl_3_, KI, NaBH_4_, and ammonia solution (28 wt.%) were used to prepare Ag and magnetite nanoparticles (NPs). Methylene blue (MB), hydrogen peroxide (30 wt.%) and phosphate buffer solutions (H_3_PO_4_/NaH_2_PO_4_) were used to evaluate the adsorption and catalytic activities of the prepared composites.

### 4.2. Preparation Methods

#### 4.2.1. Synthesis of AMPS/AA Microgel and Hydrogel

Solutions of AMPS (4.5 g, 21.7 mmol), AA (1.54g, 21.7 mmol), and MBA (0.6 g) in water (30 mL) were stirred in an ice bath. Span-85 (4.5 g) dissolved in cyclohexane (80 g) was added to the monomer solution and homogenized with homogenizer at a speed of 9500 rpm for 30 min at temperature 10 °C. The reaction mixture was bubbled with nitrogen for 15 min and the temperature rose up to 40 °C for 15 min. APS (124 mg, 0.54 mmol) solubilized in 5 mL H_2_O was added to the reaction mixture during 30 min interval. The temperature of reaction increased to 60 °C and maintained for 4 h under nitrogen flow. The polymerization was inhibited after adding methanol (5 mL) with cooling down the reaction mixture. The AMPS/AA microgel was separated from the reaction solution using ultracentrifuge at 12,000 rpm for 30 min. The solid particles were washed several times using acetone and dried overnight at 35 °C in a vacuum oven. The AMPS/AA microgel solution was subjected to the dialysis membrane for 7 days to remove the Span-85.

The AMPs/AA hydrogels were prepared with the same recipe in the absence of Span-85, and hexane by solution radical crosslinking copolymerization.

#### 4.2.2. Synthesis of AMPS/AA Nanocomposites

Solutions of KI (0.33 g dissolved in 1.5 mL of distilled water) and ferric chloride (1 g dissolved in 30 mL of distilled water) were mixed for 1 h under nitrogen gas atmosphere. The filtrate of the reaction mixture was mixed and stirred with dry AMPS/AA hydrogel or microgel (0.5 g). The AMPS/AA microgel or hydrogels rinsed into iron cation solution to reach the equilibrium. The swelled gel was stirred with 30 mL ammonia solution (25%) and stirred for 3 h to produce AMPS/AA-Fe_3_O_4_ composites.

The AMPS/AA-Ag nanocomposite was prepared by immersion of crosslinked AMPS/AA (0.5 g) hydrogel or microgel into AgNO_3_ solution (100 mL; 0.1 mM) for 24 h. The swelled gels were treated with NaBH_4_ (1 M; 100 mL) under vigorous stirring at 40 °C. The external surface of brown solids was washed several times with water and ethanol until the filtrate becomes clear followed by drying in vacuum oven at 30 °C.

### 4.3. Characterization

The surface morphology of AMPS/AA composites were examined by transmittance and scanning electron microscope (TEM and SEM JEOL JEM-2100; JEOL, Tokyo, Japan) an acceleration voltage ranged from 150 to 200 kV). Laser Zeta meter Malvern Instruments (Model Zetasizer 2000; Malvern Instruments, Malvern, UK) was used to determine the surface charges and particle size diameter of AMPS/AA microgel composites. The thermal stability and Ag or Fe_3_O_4_ contents were evaluated by thermogravimetric analysis (TGA; TGA-50; Shimadzu Co, Canby, OR, USA) using nitrogen atmosphere under flow rate 50 mL/min at a heating rate of 10 °C. min^−1^. XRD analysis was carried out by using an X-ray diffractometer (X’Pert, Philips, Amsterdam, Netherlands) with Cu K*α* radiation of wavelength 1.54 Å, operating at a voltage of 40 kV and a current of 40 mA at a rate of 2° min^−1^ and in the range 2*θ* = 0–100°. Vibrating sample magnetometer (VSM; USALDJ9600-1; LDJ Electronics, MI, USA) was used to evaluate the magnetic properties of the AMPS/AA–Fe_3_O_4_ composites. A UV−visible spectrophotometer (Shimadzu UV-1208 model; Canby, OR, USA) was used to determine the MB dye concentrations in aqueous solution at wavelength λ_max_ equals 662 nm.

### 4.4. Application of AMPS/AA Hydrogel Composites as Adsorbents

The removal efficiencies (E (%), adsorption capacity q_e_ (mg g^−1^) and the adsorption kinetics of MB onto AMPS/AA hydrogel composites were evaluated using different concentrations of MB in 50 mL of aqueous solution at 25 °C. The E (%) and q_e_ were calculated as
q_e_ = (C_o_ − C_e_) × V∕m(4)
E (%) = (C_o_ − C_e_) × 100∕C_o_(5)
where C_o_, C_e_, V, and m are the liquid phase concentrations of dye initially and at equilibrium (mg L^−1^), the volume (L) of the solution, and the mass of adsorbent used (g), respectively. The MB concentrations were measured from the absorbance of the peak at 662 nm using a UV-visible spectrophotometer.

The MB was desorbed from the AMPS/AA composites by treatment with ethanol and 0.5 mol L^−1^ HCl and neutralization with 0.1 mol L^−1^ NaOH aqueous solutions.

### 4.5. Application of AMPS/AA Composites as Catalyst

The AMPS/AA-Fe_3_O_4_ microgel composite was evaluated for Fenton oxidation of MB. In this respect, AMPS/AA-Fe_3_O_4_ (2.5 g L^−1^) was suspended into water (100 mL containing 100 mg L^−1^) using ultrasonication. HCl (0.5 mL of 6 M) and hydrogen peroxide (4 mL of 30%) were added to the AMPS/AA-Fe_3_O_4_ microgel suspension. The flask containing MB solution was closed and stirred with a magnetic stirrer over a time period and samples (2 mL) were withdrawn after separation AMPS/AA-Fe_3_O_4_ particles with an external magnet every 2 min from the mixture to determine the residual concentration of MB at 665 nm. The blank solution with absence of AMPS/AA-Fe_3_O_4_ was used to compare the MB degradation in the absence and presence of AMPS/AA-Fe_3_O_4_ microgel nanocomposites.

## 5. Conclusions

The crosslinked AMPS/AA macro- or microgel composites based on Ag and magnetite nanocomposites were prepared for application as a water pollutant adsorbent and catalyst for water purification. The data reveal that the Ag NPs are uniformly highly dispersed into AMPS/AA networks more than Fe_3_O_4_ nanoparticles and confirmed the high affinity of iron cations to combine with the AMPS/AA networks. The TGA data indicate higher contents of Ag and Fe_3_O_4_ NPs were embedded in AMPS/AA microgel composites than their hydrogel composites. The in situ technique produced core/shell morphologies for AMPS/AA-Fe_3_O_4_ microgel composites. The rough surfaces observed in case of AMPS/AA-Ag microgel demonstrate the formation of dense AMPS/AA crosslinked networks on the periphery of their microgel particles. The low negative surface charge values of AMPS/AA microgel nanocomposites in basic and acidic medium suggest that the formation of hard-crosslinked shell and weakly crosslinked core prevent the deprotonation of sulfonic and carboxylic groups of AMPS/AA network even in basic aqueous solutions. The adsorption rates of hydrogel composites arranged as AMPS/AA-Ag > AMPS/AA-Fe_3_O_4_ > AMPS/AA and suggest that the physicochemical interaction mechanism is more favorable. Moreover, the rough regular and spherical Ag NPs, beside the nanometer scale and relatively high specific surface area, are also responsible for higher MB adsorption efficiency with AMPS/AA-Ag nanocomposites. It was also confirmed that the presence of AMPS/AA-Fe_3_O_4_ microgel composites lead to complete degradation of MB without the formation of intermediates during 35 min. The high degradation rate of the AMPS/AA-Fe_3_O_4_ microgel nanocomposite correlated well with the contents of Fe_3_O_4_ (25 wt.%) having large specific surface area, good dispersion of the AMPS/AA-Fe_3_O_4_ microgel nanocomposite in their aqueous solutions, and their relatively low crystallinity due to capping with AMPS/AA amorphous microgel. The catalytic characteristics of AMPS/AA-Fe_3_O_4_ are attributed to the adsorption of MB by AMPS/AA networks followed by oxidative degradation on the outer surface of the Fe_3_O_4_ nanoparticles.

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
