# Peer review of "Hybrid Ionic Silver and Magnetite Microgels Nanocomposites for Efficient Removal of Methylene Blue"

_molecules, 2019, doi:10.3390/molecules24213867_

Round 1

Reviewer 1 Report

This manuscript reported by Atta and coworkers has shown the preparation and characterization (TEM, SEM, TGA, DLS, Zeta potentials, and magnetic properties) of AMPS/AA, AMPS/AA-Ag, AMPS/AA-Fe3O4 nanocomposites. When applying in removal of methylene blue from water, these nanocomposites displayed high adsorption capacity in a pseudo-second order kinetic model and also demonstrated complete degradation of methylene blue without the formation of organic intermediates during 35 min. with a short good were applied in adsorptive removal and degradation of methylene blue (MB) in aqueous solution. The authors did almost experiments and made discussions. Further, this paper is somewhat interesting to the readers of Molecules, and thus is recommended to be publishing. However, there are scientific and technical concerns should be considered prior to publication. 1. XRPD and IR measurements are very powerful for the characterization of AMPS/AA, AMPS/AA-Ag, AMPS/AA-Fe3O4 nanocomposites, which are lacked in the manuscript. Providing XRPD and IR data is required. 2. Please rearrange Figure 1c and Figure 1d. 3. Table 1, please include the analysis data from the first decomposition stage. 4. The size of Fe3O4 nanoparticles should be mentioned. 5. Please figure out the reusability of AMPS/AA, AMPS/AA-Ag, AMPS/AA-Fe3O4 nanocomposites in adsorption of methylene blue. 6. The title focuses on the catalytic degradation of methylene blue, which is worked by using AMPS/AA-Fe3O4 nanocomposites and only stated in a very small portion in this manuscript. So one suggestion is re-titled for this manuscript. 7. The format of reference should be noticed.

Author Response

Reviewer1  
Comments and Suggestions for Authors

XRPD and IR measurements are very powerful for the characterization of AMPS/AA, AMPS/AA-Ag, AMPS/AA-Fe3O4 nanocomposites, which are lacked in the manuscript. Providing XRPD and IR data is required.

Answer:

The XRD data added and represented in Figure 4a-d) which confirm the interactions of nanoparticles with AMPS/AA network. The AMPS/AA hydrogels diffractograms (Figure 4a and c) show semi-crystalline peak at 2-theta from 15 to 20o that affect the appearance of the Ag or Fe3O4 nanoparticles peaks. These data elucidate the interaction of Ag or Fe3O4 nanoparticles with AMPS/AA networks [20,31].

XRD and IR measurements were added and discussed

Please rearrange Figure 1c and Figure 1d.

Answer:

Figures 1c and 1d were rearranged

Table 1, please include the analysis data from the first decomposition stage.

Answer:

The analysis data from the first decomposition stage were added to Table 1.

The size of Fe3O4 nanoparticles should be mentioned. 5.

Answer:

The size of Fe3O4 nanoparticles was added

Please figure out the reusability of AMPS/AA, AMPS/AA-Ag, AMPS/AA-Fe3O4 nanocomposites in adsorption of methylene blue.

Answer:

The reusability of AMPS/AA, AMPS/AA-Ag, AMPS/AA-Fe3O4 composites was figured out (Figure 10).

The title focuses on the catalytic degradation of methylene blue, which is worked by using AMPS/AA-Fe3O4 nanocomposites and only stated in a very small portion in this manuscript. So one suggestion is re-titled for this manuscript. 7. The format of reference should be noticed.

Answer:

The title was modified to be “Hybrid Ionic Silver and Magnetite Microgels Nanocomposites for Efficient Removal of Methylene Blue”

Reviewer 2 Report

The manuscript on silver and magnetic microgels has some interesting elements, and some results might be useful for others. However, the overall quality is very low, data have not been described and discussed well, and thus it is difficult to agree (or not) with conclusions. Therefore, I recommend to rewrite and reedit the paper before any possible consideration for publication. Some other remarks are shown below:

1)

Authors probably modified SEM and TEM images (Figures 1 and 2), and thus they did not keep original proportion (X/Y). The original images should be places, and original proportion (X:Y ) must be kept during any modifications. Moreover, scale bars should be much larger with clear definitions, the image numbers (a, b, c..) should be unified (same size, font, etc.).

2)

Some data were not even described, e.g., microscopy (75-77) – only written that images are shown in figures. Therefore, the discussion and conclusions are unclear, e.g., how authors know that Ag was uniformly dispersed? (no descriptions of TEM images)

3)

Authors should be more careful. There are many mistakes, e.g., Kinetics parameters shown for first and second order reactions in Table 4 (the unit of K is opposite – maybe all data are opposite? First part for first order and second part for second order ?)

4)

The quality of all figures is not good. It is really difficult to see/check data.

5)

There are many mistakes and errors, which must be carefully corrected, e.g.,

Punctuation: unnecessary spaces in chemical names (many in Abstract) and other places (before some words (83)), unnecessary dot (“mg.”), lack of superscript (“L-1”), Grammar mistakes (70-71, and many other parts), Some sentences are strange, and should be rewritten, e.g., last sentence of Abstract, 35-37, 75-77..(and many others).

Author Response

Reviewer 2

Authors

The manuscript on silver and magnetic microgels has some interesting elements, and some results might be useful for others. However, the overall quality is very low, data have not been described and discussed well, and thus it is difficult to agree (or not) with conclusions. Therefore, I recommend to rewrite and reedit the paper before any possible consideration for publication. Some other remarks are shown below:

Answer:

The paper was reedited, and all mistakes were corrected.

Authors probably modified SEM and TEM images (Figures 1 and 2), and thus they did not keep original proportion (X/Y). The original images should be places, and original proportion (X:Y ) must be kept during any modifications. Moreover, scale bars should be much larger with clear definitions, the image numbers (a, b, c..) should be unified (same size, font, etc.).

Answer:

The resolutions, scale bars of SEM and TEM images were enhanced. The original TEM and SEM images added and the images numbers deleted from the figures and the scale bar added which was removed to control size of images.

Some data were not even described, e.g., microscopy (75-77) – only written that images are shown in figures. Therefore, the discussion and conclusions are unclear, e.g., how authors know that Ag was uniformly dispersed? (no descriptions of TEM images)

Answer:

All results were presented in results section, e.g., microscopy (75-77), these results were discussed in details in discussion section.

Authors should be more careful. There are many mistakes, e.g., Kinetics parameters shown for first and second order reactions in Table 4 (the unit of K is opposite – maybe all data are opposite? First part for first order and second part for second order ?)

Answer:

Mistakes in entire manuscript were corrected. The units of K were corrected. The unites of K only were opposites.

The quality of all figures is not good. It is really difficult to see/check data.

Answer:

The quality of all figures was enhanced

There are many mistakes and errors, which must be carefully corrected, e.g., Punctuation: unnecessary spaces in chemical names (many in Abstract) and other places (before some words (83)), unnecessary dot (“mg.”), lack of superscript (“L-1”), Grammar mistakes (70-71, and many other parts), Some sentences are strange, and should be rewritten, e.g., last sentence of Abstract, 35-37, 75-77..(and many others).

Answer:

The entire manuscript was revised and all mistakes were corrected

Round 2

Reviewer 2 Report

Although, authors corrected some parts, there are some points, which must be improved before possible acceptance, e.g.,

1) Microscopy images must be improve - please check other good papers with SEM/TEM images - square images are recommended with original scale (X:Y) and same kind of description, i.e., style of scale bars (not different in all images) - preferable drawn by authors (not from original images) with clear scale!

2) Results part must DESCRIBE results! They can not be only describe in discussion part. "Results" means results not only that "Fig. X presents X." 

All my previous remarks must be considered/corrected carefully.

Author Response

Referring to our manuscript “molecules-623926” entitled “Hybrid Ionic Silver and Magnetite Microgels Nanocomposites for Efficient Removal of Methylene Blue”.

All comments were considered, we made all corrections according to the reviewer recommendations.

 All corrections were highlighted in red

Reviewer comments:

Although, authors corrected some parts, there are some points, which must be improved before possible acceptance, e.g.,

Microscopy images must be improve - please check other good papers with SEM/TEM images - square images are recommended with original scale (X:Y) and same kind of description, i.e., style of scale bars (not different in all images) - preferable drawn by authors (not from original images) with clear scale!

Answer

The microscopy images were improved. The scale bar for all images were uniformed.

Results part must DESCRIBE results! They can not be only describe in discussion part. "Results" means results not only that "Fig. X presents X." 

Answer

Results were described in the results section.